# Quantifying the Buttressing Contribution of Landfast Sea Ice and Melange to Crane Glacier, Antarctic Peninsula

Richard Parsons[1], Sainan Sun[1], G. Hilmar Gudmundsson[1], Jan Wuite[2], and Thomas Nagler[2]

[1]Department of Geography and Environmental Sciences, Northumbria University, Newcastle upon Tyne, NE1 8ST, UK
[2]ENVEO IT GmbH, Fürstenweg 176, A-6020 Innsbruck, Austria

**Correspondence:** Richard Parsons (richard.parsons@northumbria.ac.uk)

**Abstract.** The January 2022 disintegration of multi-year landfast sea ice in the Larsen B Embayment, Antarctic Peninsula, was closely followed by a significant acceleration of ice flow and ice-front retreat of numerous outlet glaciers. Crane Glacier was a notable example of this, with 6 km of its floating ice shelf lost to calving in the first month following the disintegration and a 3.4% increase in terminus flow speeds over the same time period. In this study we quantify for the first time the buttressing stresses that were transmitted to Crane by the ice melange at the glacier outlet using ice-flow model, Úa. We constrained our model with high resolution surface elevation profiles of the glacier and ambient melange and reconstructed the observed flow velocities by optimising the rheology rate factor throughout our model domain. This allowed us to quantify the stress regime across both the glacier and ice melange. Results showed that resistive backstresses were imparted to Crane by the ice melange with a mean buttressing ratio of $\Theta_N = 0.68$ calculated at the glacier terminus ($\Theta_N = 1$ implies no buttressing). In addition, diagnostic modelling showed an expected 19.2 kPa mean increase in extensional stress at the ice-front following the disintegration of the ice melange. This perturbation in stress likely triggered the observed rapid calving over the near terminus region, leading to the periodic loss of sections of Crane's buttressing ice shelf and thus further acceleration of ice flow in the subsequent months.

## 1 Introduction

Landfast sea ice (fast ice) is a type of stationary sea ice which is fastened to coastlines, ice shelves and outlet glaciers. Despite its potential to regulate the dynamic behaviour of adjoining glaciers (Massom et al., 2010, 2015; Arthur et al., 2021; Christie et al., 2022), gaps in knowledge relating to the distribution and extent of fast ice have led to it being overlooked in many studies (Fraser et al., 2023). Recent observations have significantly improved our understanding of trends in fast ice extent throughout Antarctica (Fraser et al., 2021), changes to which could impact upon the stability of adjoining ice shelves and outlet glaciers, therefore potentially impacting future rates of sea level rise (Massom et al., 2018; Miles et al., 2017).

Whilst the buttressing capacity of ice shelves has been assessed quantitatively in numerous studies (Fürst et al., 2016; Reese et al., 2018; Gudmundsson et al., 2023), buttressing provided by fast ice has primarily been discussed in terms of correlation between perturbations in fast ice coverage and observed changes in ice flow velocity or migration of ice tongue calving fronts. Seasonal acceleration of the Totton Ice Shelf (Greene et al., 2018) and the Parker Ice tongue (Gomez-Fell et al., 2022) have

been attributed to changes in fast ice extent, as has intermittent acceleration of the Thwaites Glacier tongue (Miles et al., 2020). However, little to no change in glacier flow velocities were reported following fast ice breakout events at Shirase Glacier (Nakamura et al., 2010, 2022). The contrasting responses reported in these studies suggest that either seasonal or local geometric factors may ultimately control the extent to which fast ice may stabilise connected glaciers (Fraser et al., 2023).

More recently, the 2022 widespread disintegration of multi-year fast ice in the Larsen B Embayment, Antarctic Peninsula,
provided an opportunity to assess the impact of instantaneous fast ice loss to multiple glaciers, aided by abundant present-day observational datasets for the region (Sun et al., 2023; Ochwat et al., 2024; Surawy-Stepney et al., 2024). Furthermore, the capacity of fast ice to buttress glaciers was brought in to question due to the similarities in the response of the regions outlet glaciers following this disintegration event and the 2002 collapse of the Larsen B Ice Shelf (Scambos et al., 2004; Rignot et al., 2004; De Rydt et al., 2015).

Following the 2002 ice-shelf collapse, Crane Glacier (Fig. 1) retreated by more than 10 km in just over two years (Needell and Holschuh, 2023) with ice flow across the grounding line increasing by up to three times over a similar period (Rignot et al., 2004). Crane's longer-term response was more complex and following the initial rapid retreat, phases of arrest and subsequent re-advance were exhibited (Needell and Holschuh, 2023; Rott et al., 2018; Wuite et al., 2015).

Sea ice formed seasonally in the Larsen B embayment each year following the 2002 ice-shelf collapse, but in 2011 the sea
ice became landfast before further developing into multi-year fast ice. Multi-year fast ice is not only thicker than seasonal formations of sea ice (i.e. first-year ice) (Arthur et al., 2021; Massom et al., 2010), but can also display enhanced mechanical strength through desalination over summer months (Fraser et al., 2023). This thicker, stronger ice allowed the trapping and bonding of fragments of partially calved icebergs in a pro-glacial melange (Ochwat et al., 2024; Surawy-Stepney et al., 2024). The thickness of the multi-year fast ice was estimated to be between 2.5 m and 4 m across the embayment (Scambos et al.,
2017), though thicknesses of magnitude of tens to hundreds of metres was reported for melange elements close to the glacier outlets (Ochwat et al., 2024). The development of multi-year fast ice seemingly aided a decade long re-advance of Crane, but its disintegration in mid-January 2022 triggered a second period of rapid retreat (Needell and Holschuh, 2023). Ice flow velocities were also affected with acceleration observed in the months following the disintegration, however a more significant increase in flow speeds over both grounded and floating ice was not observed until later in the year (Fig. 2e, Movie S2).

The similarities between the 2002 and 2022 events led to multiple studies assessing the role that the fast ice played in buttressing Larsen B's outlet glaciers, including, but not limited to, Crane glacier (Sun et al., 2023; Ochwat et al., 2024; Surawy-Stepney et al., 2024). However, conclusions on the extent to which fast ice can buttress adjoining glaciers varied across these studies. Sun et al. (2023) argued that whilst the fast ice covered the same area as the previously existing ice shelf, the buttressing provided was low as damage could more readily propagate through the fast ice due to its weaker and
thinner makeup, therefore limiting its resistive potential. This was evidenced by a lack of instantaneous acceleration at six glaciers following the disintegration and near-zero correlation between fast ice extent and glacier velocities (Sun et al., 2023). A delayed velocity response, observed 8 months later, was attributed to the retreat of Crane's ice front, rather than being directly due to the collapse of the fast ice (Sun et al., 2023). In contrast, Ochwat et al. (2024) recognised the similarities in dynamic response following the fast ice and ice shelf break-ups, including instantaneous acceleration of ice flow, to mean there were

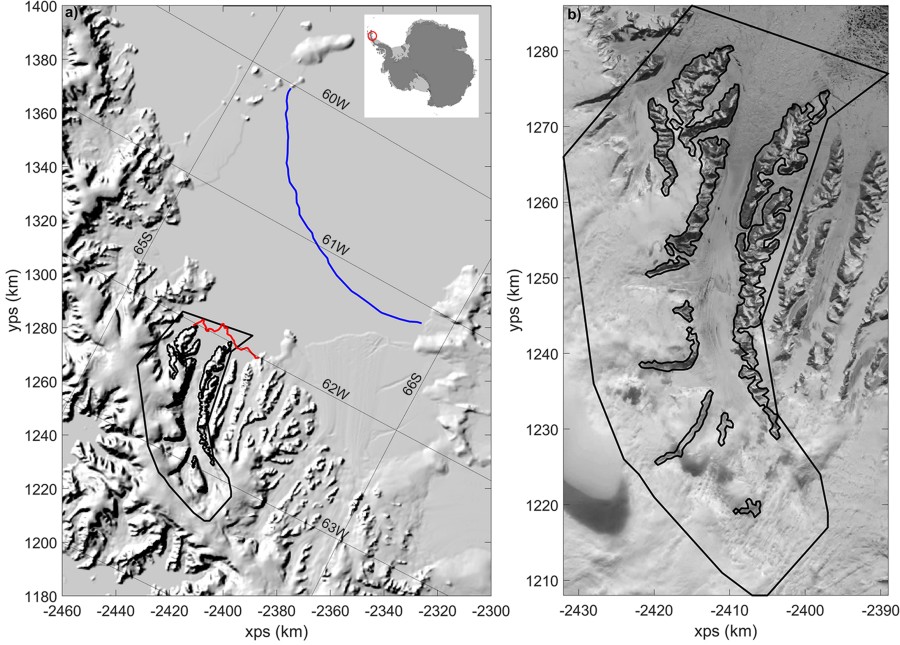

**Figure 1.** Panel a) shows an overview of the Larsen B embayment in polar stereographic projection displayed on a hill shaded relief map of the REMA mosaic for Antarctica (Howat et al., 2019). Latitude and longitude lines are labelled for reference. The location of Larsen B is illustrated by the red ring on the subset outline of Antarctica (source: BedMachine v3 (Morlighem, 2022)). The black line outlines the extent of the model domain. The blue line denotes the approximate extent of fast ice in the months prior to its disintegration and the red line shows the fast ice and melange extent after disintegration with coordinates extracted from LandSat 8 imagery (24/02/2022). Panel b) shows a close up view of Crane Glacier captured by LandSat 9 (13/12/2021) with the black line showing the extent of the model domain.

parallels in the buttressing provided by the two different ice masses. The destabilisation and calving of floating ice tongues in different outlet glaciers in Larsen B were therefore attributed to the loss of buttressing from the fast ice (Ochwat et al., 2024). Surawy-Stepney et al. (2024) used an ice-sheet model to estimate the buttressing stresses of an idealised distribution of fast ice on Larsen B's outlet glaciers. The authors first optimised the rate factor of the glaciers to reproduce the observed velocity field without the inclusion of fast ice in the model, then added varying thicknesses of fast ice to the embayment region
of the domain to calculate the resulting change in ice-flow speed. The study concluded that whilst the fast ice affected the dynamic behaviour of floating ice in different glacier outlets, this was not a result of direct buttressing in the same context as is understood for ice shelves. Instead, the impacts that fast ice coverage had on calving behaviour and glacier flow velocities were attributed to secondary processes including ocean swell attenuation (Teder et al., 2022; Christie et al., 2022) and bonding of melange in areas downstream of glacier termini (Robel, 2017). The potential issue of this approach lies in the initialisation
process: by solving the optimisation problem without including fast ice, the buttressing stresses from the fast ice, if any, have been compensated by adjusting the rate factor of the glacier.

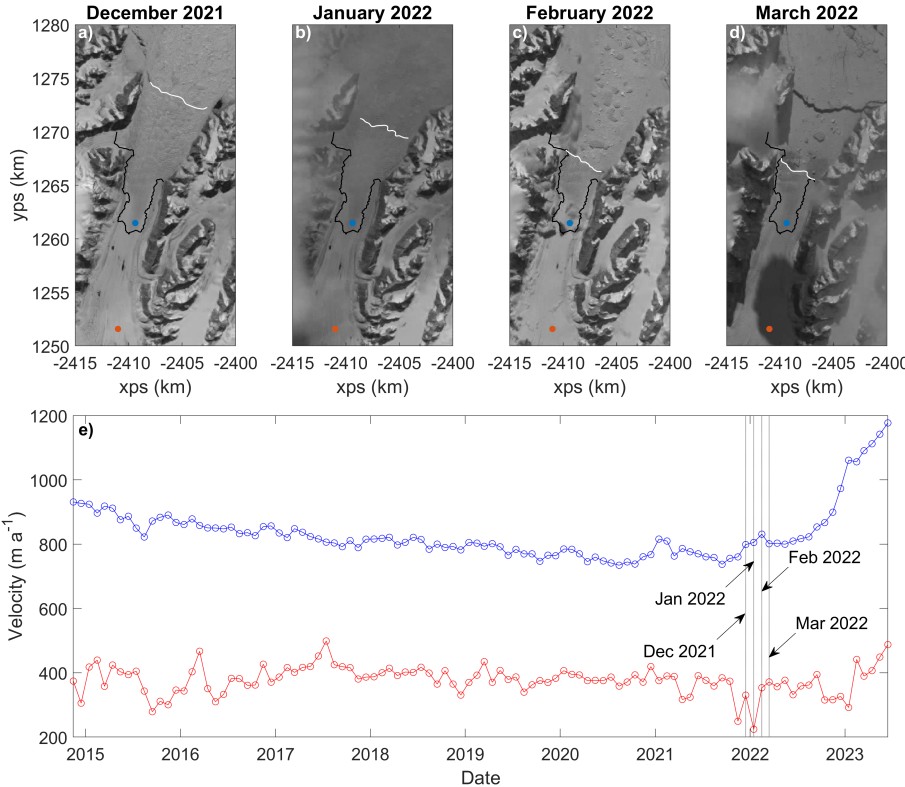

**Figure 2.** Top row: Crane's outlet in a) December 2021 and b) January 2022, c) February 2022, d) March 2022 (acquisition dates provided in Table 1), cropped from Landsat 8 and 9 satellite images. The fast ice was still intact in December 2021 and a degree of ice melange remained in the outlet of the fjord in the months following the fast ice disintegration. The black line shows the modelled grounding line position and the white line shows the digitised terminus location. The coloured dots are locations from which velocity data was extracted by ENVEO et al. (2021). Monthly averaged ice flow velocities from 2015-2023 (ENVEO et al., 2021) are shown in panel e). The blue and red lines represent the flow velocities at the corresponding blue and red location markers shown in the top row images with the black vertical lines corresponding to the dates of panels a) - d).

Here we seek to quantify for the first time the buttressing stresses that were imparted to Crane Glacier by the observed mixture of fast ice and pro-glacial melange. For this study we use ice-sheet model, Úa, with a computational domain which includes the glacier as well as ambient fast ice and pro-glacial melange. We quantify buttressing using the buttressing ratio (Gudmundsson, 2013; Schoof, 2007) and further assess the change in the near-terminus stress field following the fast ice disintegration, which may have triggered retreat of Crane's ice-front. Finally, we test the robustness of the ice-sheet model in determining the resistive stresses imparted to Crane by investigating the sensitivity of our results to changes in the prescribed input thickness of fast ice and melange elements. This is to improve confidence in employing our methodology in scenarios where uncertainty in ice thickness measurements is of concern.

## 2 Datasets

In order to configure a model geometry representative of Crane Glacier prior to the disintegration of fast ice in the Larsen B embayment, we utilised a range of REMA strip DEMs (Howat et al., 2019), timestamped between 30th September 2020 and 16th January 2022. The strips are defined at 2 m spatial resolution and have absolute error of $\pm 2$ m in horizontal and vertical planes. To create a continuous surface elevation profile across our model domain, each strip was added in turn and where coordinates overlapped between strips, the most recent strip was used. As elevations at intersecting locations did not deviate beyond the associated error range, no further corrections were applied between intersecting strips. The near-terminus region and melange filled outlet of Crane were populated by the strip dated 16th January 2022. The REMA strip DEMs are referenced to the WGS64 ellipsoid and were corrected for the geoid using values from BedMachine (Morlighem, 2022; Morlighem et al., 2020) interpolated at each nodal point of our model domain.

The bedrock elevation was defined using the Huss and Farinotti (2014) bedrock DEM, who used a mass continuity approach informed by NASA's Operation IceBridge and ground based measurements to define a bedrock dataset across the Antarctic Peninsula at 100 m resolution. This dataset was merged with multibeam swath bathymetry data (Rebesco et al., 2014) which provided direct measurements of bedrock elevations in the vicinity of Crane's outlet and grounding line. However, due to uncertainty in the bedrock profile and recent observations suggesting that Crane's grounding line lay further downstream prior to the fast ice disintegration (Wallis et al., 2024), we performed an additional sensitivity experiment considering the shallower Bedmachine bed elevation data (Morlighem, 2022; Morlighem et al., 2020) to ensure that our conclusions would not be impacted by uncertainty in the estimated bed topography (Fig. S4).

The ice thicknesses across regions of floating ice were calculated from the flotation criterion considering the input surface elevations (Howat et al., 2019) and uniform ice and ocean densities of 917kg m$^{-3}$ and 1030kg m$^{-3}$ respectively. The grounding line is defined in our model where the calculated submerged ice thickness reaches the prescribed bedrock elevation (Huss and Farinotti, 2014).

Monthly averaged ice velocity maps at 200 m grid spacing were derived from successive Sentinel-1 IW SLC image pairs (2014-2023) using a combination of coherent and incoherent offset tracking techniques (ENVEO et al., 2021; Nagler et al., 2021, 2015).

## 3 Methodology

### 3.1 Experiment Design

We performed a series of numerical simulations to quantify the buttressing stresses provided to Crane Glacier by the fast ice and pro-glacial melange. The simulations were performed in two steps, an overview of which is given below with further details described in the following subsections.

In the first step, the rate factor, $A$ in Glen's flow law, was estimated through inversion of measured velocities, allowing us to reconstruct the stress field throughout Crane and the ambient ice melange. We used satellite imagery to extract the location

**Table 1.** The terminus ID's referred to throughout the study represent the terminus locations extracted from satellite imagery with acquisition dates shown below. LandSat imagery was obtained courtesy of the U.S. Geological Survey

| Terminus ID | Date of Acquisition | Satellite |
|---|---|---|
| December 2021 | 13/12/2021 | LandSat 9 |
| January 2022 | 21/01/2022 | LandSat 9 |
| February 2022 | 24/02/2022 | LandSat 8 |
| March 2022 | 19/03/2022 | LandSat 9 |

of the calving front prior to the fast ice disintegration and calculated the normal resistive stress at these coordinates in order to quantify the buttressing provided by the fast ice and melange. We made the same calculations at the location of the terminus in the days immediately after the fast ice disintegration and again at locations that the ice-front had retreated to in the following months (Table 1). These terminus locations were considered in order to compare the buttressing stresses provided by the fast ice and melange with those provided by the portions of Crane's floating ice shelf which were lost to calving in the months following the fast ice disintegration.

In the second step, we investigated the expected change in the near terminus stress regime following the fast ice disintegration. After having inverted for $A$ and ensured that modelled ice velocities were in good agreement with observations, we perturbed the model in a diagnostic simulation by removing the ice melange from the domain and assessed the instantaneous change in stress regime.

## 3.2 Model Setup

We used the ice-flow model, Úa (Gudmundsson, 2020), which solves the governing equations of ice dynamics using the SSA approximation (Morland, 1987; MacAyeal, 1989). The model uses a 2D vertically integrated approach which allowed us to assess the stress distribution throughout the domain in lateral and transverse directions, enabling an investigation into buttressing from the fast ice and melange where effects of lateral drag from the margins of the fjord are captured.

The model domain included the entirety of Crane, along with all in-flowing tributary regions of the glacier and extended downstream from the outlet of Crane's fjord (Figure 1). The location of rock outcrops were defined using Landsat imagery and holes in the mesh were placed at these locations where areas of thin ice and high strain rates may have caused numerical difficulties for the ice flow model (De Rydt et al., 2015). Boundary conditions at the edges of the domain were fixed to ice flow velocities from observational data (ENVEO et al., 2021). A zero flow condition was imposed at the interior boundaries.

The finite element mesh was refined to a minimum resolution of 100 m over areas of fast ice and melange and regions close to the terminus and grounding line positions of the glacier. To reduce computational cost, a coarser resolution was employed further upstream and varied based on flow velocities up to a maximum resolution of 2.5 km. A total of 36268 elements with mean size of 194.2 m made up the model domain.

### 3.3 Inversion

Model parameters for basal slipperiness ($C$) and ice rheology (rate factor, $A$) were determined through an inversion process (MacAyeal, 1993) based on ice velocity measurements following a commonly used methodology (e.g. Hill et al., 2018; Barnes et al., 2020; Sun and Gudmundsson, 2023). Ice rheology is assumed to follow Glen's Flow Law (Glen, 1955) with stress exponent, $n = 3$, and basal sliding is assumed to follow Weertman's sliding law (Weertman, 1957), with stress exponent, $m = 3$.

Prior values for A and C were chosen based on assumed temperature and flow velocity respectively, before iteratively updating these values to minimize the cost function, $J = I + R$. Here, $I$, represents the misfit between modelled and observed velocities and, $R$, is a regularization parameter, for which we use Tikhonov regularisation of the form

$$R = \frac{1}{2\mathcal{A}} \int \left[ \gamma_{sA}^2 \left( \nabla \log_{10} \left( \frac{A}{\tilde{A}} \right) \right)^2 + \gamma_{sC}^2 \left( \nabla \log_{10} \left( \frac{C}{\tilde{C}} \right) \right)^2 + \gamma_{aA}^2 \left( \nabla \log_{10} \left( \frac{A}{\tilde{A}} \right) \right)^2 + \gamma_{aC}^2 \left( \nabla \log_{10} \left( \frac{C}{\tilde{C}} \right) \right)^2 \right] d\mathcal{A} \quad (1)$$

where $\mathcal{A}$ is the domain area, $\tilde{A}$ and $\tilde{C}$ are prior estimates for $A$ and $C$ and $\gamma_s$ and $\gamma_a$ are regularistion parameters penalising slope and amplitude respectively. We performed L-curve analysis (Hansen, 1992) to optimize the regularization parameters, iterating until convergence.

The inversion was based on monthly averaged velocity measurements from November 2021. This negated the possibility of including any contamination which may be present in the January 2022 dataset which partly accounts for time periods both prior to and following the fast ice disintegration. Monthly averaged velocities from December 2021 were discounted due to high errors associated with the velocity measurements downstream of the terminus location, however results from sensitivity experiments show the choice of velocity dataset used in the inversion has little impact on the findings of the study (Fig. S6).

### 3.4 Diagnostic Modelling

After having inverted for $A$ and ensured that modelled ice velocities were in good agreement with observations, we performed a diagnostic simulation in order to assess the instantaneous change in the near-terminus stress regime following the disintegration of the fast ice. We perturbed the model by removing all ice from the model domain downstream of the December 2021 terminus location (Table 1). It is noted that for numerical reasons, removing the fast ice and melange actually involves replacing the defined thickness from the REMA strip DEMs (Howat et al., 2019) with a nominal 0.1 m thickness, which is suitably thin so as not to impact the upstream flow of ice.

Prior to running the diagnostic simulation, the boundary conditions at the domain's flow outlet were changed to a stress-free condition.

### 3.5 Buttressing Quantification

Buttressing is commonly quantified by determining the normal resistive stress imparted by an ice shelf at the grounding line, $R_N$, and comparing this to the hypothetical ocean pressure which would be exerted at the same location in the absence of an

ice shelf, $R_o$ (Schoof, 2007; Gudmundsson, 2013). Here we apply the same methodology, instead considering the vertically integrated normal resistive stress imparted by the fast ice and melange elements at the glacier terminus.

$$R_o = \frac{1}{2}\rho_i \left(1 - \frac{\rho_i}{\rho_o}\right) gh \tag{2}$$

$$R_N = \boldsymbol{n}^\mathsf{T} \cdot (\boldsymbol{R}\boldsymbol{n}) \tag{3}$$

$$R = \begin{bmatrix} 2\tau_{xx} + \tau_{yy} & \tau_{xy} \\ \tau_{xy} & 2\tau_{yy} + \tau_{xx} \end{bmatrix} \tag{4}$$

where $g$ is the gravitational constant, $h$ is ice thickness, $\boldsymbol{n}$ is the unit vector normal to the terminus, $R$ is the resistive stress tensor and $\tau_{ij}$ are components of the deviatoric stress tensor computed in the model inversion. Values of 917kg m$^{-3}$ and
175 1030kg m$^{-3}$ are considered for ice density, $\rho_i$, and sea water density, $\rho_o$, respectively.

The buttressing ratio is given by $\Theta_N$,

$$\Theta_N = \frac{R_N}{R_o} \tag{5}$$

The scenario of no buttressing is given by $\Theta_N = 1$, as in this example the resistive stress imparted by the fast ice and melange normal to the terminus would be equal to that of the vertically integrated ocean pressure at the same location. $\Theta_N < 1$ shows
there are buttressing stresses resisting ice flow, with decreasing values corresponding to increasing resistive stress. Where $\Theta_N > 1$, resistive stresses are less than in the hypothetical ice free scenario, indicating that ice is being pulled downstream. Where $\Theta_N < 0$, deviatoric stress normal to the terminus is compressive, otherwise it is tensile.

### 3.6 Fast Ice and Melange Thickness Sensitivity

The inversion process provides an optimised estimate of the ice rheology over the glacier, fast ice and melange areas of the
185 model domain. Crucially, this process accounts for the discontinuous nature of the melange by returning a varied $A$ (rate factor) field, derived from the measured velocities and dependent upon the input ice thicknesses by solving the conservation of momentum equations with SSA approximation,

$$\partial_x(h(2\tau_{xx} + \tau_{yy})) + \partial_y(h\tau_{xy}) - t_{bx} = \rho_i gh\partial_x s \tag{6}$$

$$\partial_y(h(2\tau_{yy} + \tau_{xx})) + \partial_x(h\tau_{xy}) - t_{by} = \rho_i gh\partial_y s \tag{7}$$

where $h$ is ice thickness, $t_{bh}$ is the horizontal components of the basal stress vector (equal to zero in floating regions), $\rho_i$ is ice density, $g$ is the gravitational constant, $s$ is the ice surface elevation and $\tau_{ij}$ are components of the deviatoric stress tensor calculated using Glen's Flow Law,

$$\dot{\varepsilon}_{ij} = A\tau^{n-1}\tau_{ij} \tag{8}$$

where $\dot{\varepsilon}_{ij}$ are components of the strain rate tensor, $A$ is the rate factor, $\tau$ is the second invariant of the deviatoric stress tensor, $\tau_{ij}$, and $n$ is the stress exponent.

The product of rate factor and ice thickness are found in the left-hand side of the momentum equations (Eqns. 6 and 7), with ice thickness only appearing separately from the rate factor in the body force component on the right-hand side of the equations. We therefore tested the hypothesis that uncertainty in fast ice and melange thickness measurements are accounted for in the inversion, as the resulting ice rheology is described not solely by the rate factor, rather by the product of $A \cdot h$.

We performed a series of sensitivity experiments by adjusting the prescribed surface elevation of fast ice and melange elements downstream of the December 2021 terminus in increments of $\pm 10\%$ up to a maximum $40\%$ change from the original surface elevations (Howat et al., 2019). In regions of thin ice, the minimum defined surface elevation was limited to 0.1 m.

For each adjusted ice thickness configuration, we performed a new model inversion. Using the updated rheology rate factor fields, we calculated the buttressing ratio at the terminus location in each sensitivity case. Our hypothesis would be proven if the calculated buttressing ratios were not affected by the adjustment in the input surface elevation of fast ice and melange. Results from these sensitivity experiments are presented in Section 4.3.

## 4 Results

### 4.1 Buttressing Ratio

High spatial variability in the calculated buttressing ratios are seen at the December 2021 and January 2022 terminus positions (Figure 3). Compression is seen in the central regions (red bubbles, Figure 3) and the majority of buttressing ratios lie below 1, showing that resistive backstresses are imparted by the fast ice and melange. Buttressing ratios exceed 1 in some cases close to the edges of the fjord indicating that ice at the margins is being pulled downstream. Mean buttressing ratios of 0.68 and 0.65 were found across the December 2021 and January 2022 terminus locations respectively with values between 0.8 and 0.9 occurring most frequently along these ice fronts.

The February 2022 and March 2022 terminus positions show a more consistent grouping of buttressing ratios with the majority of calculated $\Theta_N$ values lying between 0.6 and 0.7 (Figure 3). Buttressing ratios do not exceed 1 at any point with mean values of 0.66 and 0.60 calculated along the February 2022 and March 2022 ice-fronts respectively.

### 4.2 Change in Terminus Stress Regime

We compared Crane's stress regime from our inversion, representing the stress field in the glacier with fast ice still intact, against that from the diagnostic simulation in which fast ice and melange was removed downstream of the December 2021

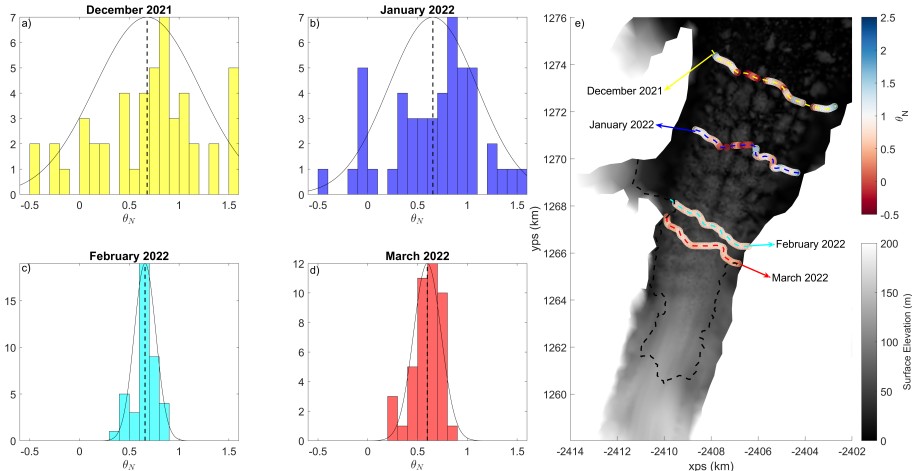

**Figure 3.** The buttressing ratio ($\Theta_N$, Eqn. 5) was defined at 100 m intervals along terminus positions derived from Landsat imagery representing the ice front in December 2021, January 2022, February 2022 and March 2022 (Table 1). Histograms of the buttressing ratios at these intervals along each month's terminus position are shown in panels a) - d) with a normally distributed probability density function overlain in black. The vertical black dashed lines show the mean buttressing ratio. Panel e) shows the buttressing ratios in coloured bubbles along the terminus location with yellow, blue, cyan and red dashed lines showing the terminus location in December 2021, January 2022, February 2022 and March 2022 respectively, plotted on top of a surface elevation map of Crane's outlet region. The dashed black line shows the grounding line.

terminus. The removal of melange elements from the domain led to an increase in extensional stresses in Crane's floating ice shelf running perpendicular to the ice front (Fig. 4). The magnitude of this change increased with closer proximity to the ice front where maximum and mean modelled increases in extensional stresses were found to be 70.8 kPa and 19.2 kPa respectively. These values can be compared to the modelled basal drag values approaching the grounding line which have an average of 64.4 kPa.

### 4.3 Sensitivity of Results to Fast Ice and Melange Thickness

The buttressing ratios calculated across the terminus remain consistent across each sensitivity experiment with the spatial distribution of areas of high and low buttressing unchanged by the input ice thickness (Fig. 5). Greater resistive stresses are seen in the central region of the ice-front with areas of negative buttressing ($\Theta_N > 1$) seen at the margins.

A 14.3% increase in the mean value of $\Theta_N$ was calculated with ice thicknesses downstream of the terminus location reduced by 40% (the thinnest thickness distribution assessed). In all other sensitivity experiments, the mean value of $\Theta_N$ differed by a maximum of 6.6% from the reference configuration. No correlation between the percentage change in modelled thickness of ice melange and the deviation of mean buttressing ratio from the reference case results is observed. Uncertainty in the ice thickness profile therefore does not significantly affect our results.

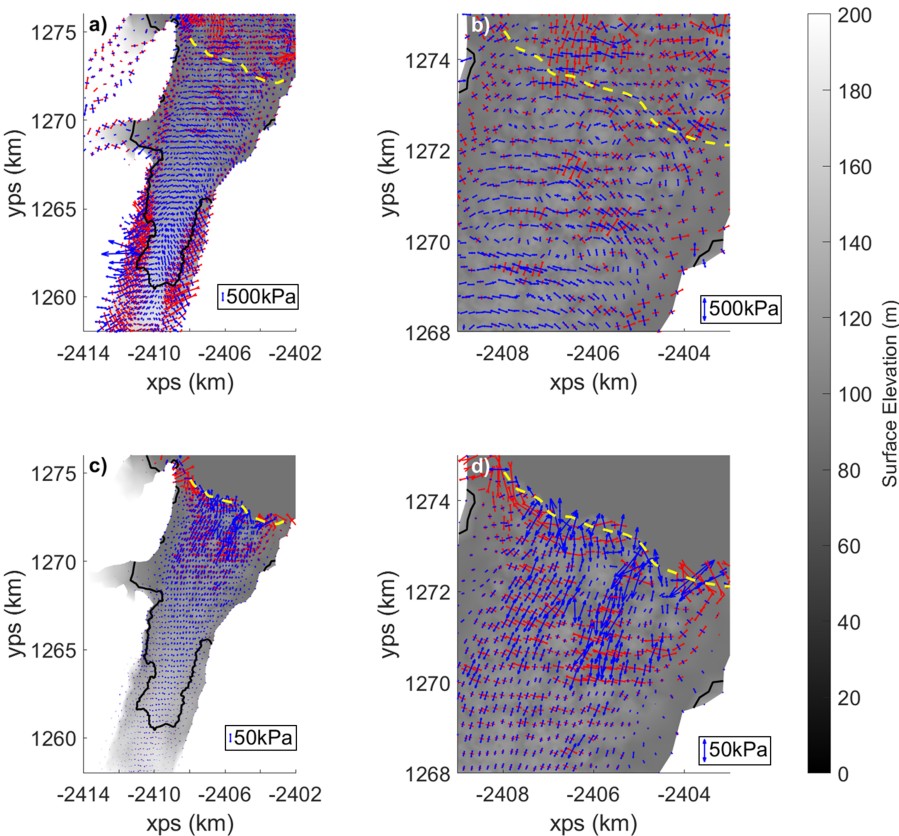

**Figure 4.** Panel a) shows the principal deviatoric stress distribution in the near-terminus region of Crane determined from the inversion with fast ice and melange included in the domain. Extensional and compressive stresses are shown by the blue and red arrows respectively. A zoomed in view of this stress distribution at the terminus is shown in panel b). Panel c) shows the change in the principle deviatoric stresses following the removal of fast ice and melange downstream of the December 2021 terminus, determined in the diagnostic model run. A zoomed in view of this stress distribution at the terminus is shown in panel d). The maximum and mean increases in the 1st principle stress along the terminus are 70.8 kPa and 19.2 kPa respectively. Stress distributions in each panel are overlain on the ice surface elevation profile with the grounding line and terminus positions illustrated by the solid black and dashed yellow lines respectively.

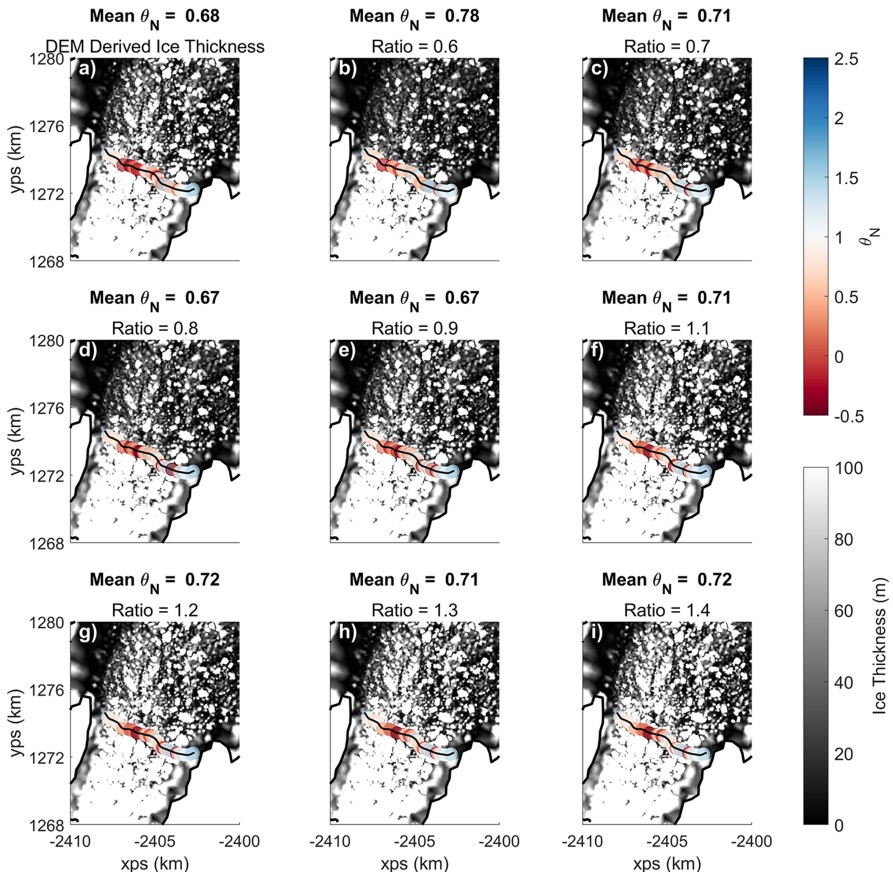

**Figure 5.** The buttressing ratios calculated at the December 2021 terminus location with ice melange represented by varying thicknesses in the model. Updated model inversions were performed separately for each configuration. In panel a), the fast ice and melange thicknesses downstream of the terminus were defined using the REMA strip DEMs as per the initial intact model configuration. In panels b) - i), ice thicknesses downstream of the terminus were adjusted from the DEM derived thickness by the ratio labelled on each panel. The solid black line shows the terminus location with buttressing ratios along this line shown by coloured bubbles at 100 m intervals.

## 5 Discussion

### 5.1 Buttressing Contribution of Fast Ice and Melange

The buttressing stresses imparted by multi-year fast ice to Larsen B's outlet glaciers has previously been assessed through interpretation of observational data (Sun et al., 2023; Ochwat et al., 2024), as well as in an idealised modelling setup assuming uniform thickness of fast ice across the Larsen B embayment (Surawy-Stepney et al., 2024). As no consensus as to the significance of the buttressing provided by the fast ice was reached between these studies, we sought to quantify this buttressing for the first time using a numerical modelling setup which captured the realistic profile of pro-glacial melange in Crane's immediate outlet. With a mean buttressing ratio of 0.68 calculated at the glacier terminus (Fig. 3), our results show that this combination of fast ice and pro-glacial melange provided significant resistive backstress to Crane.

Negative buttressing ratios, which correspond to areas of compressive stress, were found in some locations across the December 2021 and January 2022 ice fronts (Fig. 3). Examination of satellite imagery (Fig. 2a) and the surface elevation profile of the terminus region (Howat et al., 2019) highlights the discontinuous nature of the damaged ice shelf and melange in the transition between glacier and fast ice. Fragments of calved icebergs were trapped by the fast ice, preventing transport away from the terminus and making the definition of an exact terminus location difficult to achieve. When we consider that the model reduces the resolution over this region of high variability in ice thickness to a 100 m grid, we anticipate that compression is found in regions of the model which are influenced by crevassing and fragmentation of the ice. These areas are unlikely to impart such high resistive stresses as the corresponding buttressing ratios suggest, which would result in the mean ratios being artificially shifted to a lower amount. If we instead consider the mode values calculated across the December 2021 and January 2022 terminus locations, we find buttressing ratios between 0.8 and 0.9 (Fig. 3). These values may therefore be more representative of the buttressing provided by the ice melange than the mean values reported above. Despite this, the mode values still support the conclusion that buttressing was imparted by the melange prior to its disintegration and that these buttressing stresses are lower than those provided by the ice shelf further upstream (Fig. 3).

The buttressing ratios presented above are subject to a degree of uncertainty due to the difficulty in defining the exact location where the transition between glacier and melange exists. By assessing the normal resistive stresses at 100 m intervals along the defined terminus location, we aimed to minimise this uncertainty by capturing the variability in ice thickness and rheology that may exist between melange elements. We have not tested the impact on the results that would be caused by changes to the defined terminus coordinates, however similar variability in the thickness and rheology between melange elements would be expected if the terminus location was moved elsewhere in the transition zone between the damaged ice shelf and melange region.

We note that further uncertainty may exist, arising from uncertainties in the datasets used in this study, as well as being caused numerically. We tested the sensitivity of results to the bed topography considered in the model geometry (Fig. S4) and also to the velocity field considered in the inversion (Fig. S6) to demonstrate the robustness of these results. We discuss the sensitivity of the results to the prescribed thickness of the fast ice and melange in Section 5.3.

## 5.2 Crane's Response to the Fast Ice Disintegration

The quantification of buttressing from both the ice melange and the floating ice shelf (Fig. 3) provides insight into the observed response of Crane to the disintegration of Larsen B's landfast sea ice. Observational data shows 1) an acceleration of ice flow speed and 2) rapid calving of Crane's floating ice shelf.

The flow speed at the terminus increased by an average of 3.4% in the first month following the fast ice disintegration compared to when the fast ice was intact (Movie S2). A smaller increase of 0.5% across the grounding line was observed over

275 the same time period. A fluctuation between increase and decrease in flow speed across the grounding line was observed until August 2022 with average flow speeds typically within 1% of the pre-disintegration velocities, however acceleration at the terminus region reached a maximum of 13.6% over this period, suggesting that the mechanical support supplied to the glacier by the fast ice and melange played a greater role in restriction of ice flow at the terminus than further upstream at the grounding line.

Eight months later, in September 2022, increases in flow speed at the terminus and grounding line locations had increased to 14.10% and 7.54% respectively, with further increases observed in the following months (Fig. 2e, Movie S2). Sun et al. (2023) argued that such a delay in the velocity response was due to the retreat of Crane's ice front, rather than being directly due to the disintegration of the fast ice. However, our results suggest that these phenomena are connected and ultimately caused by the fast ice disintegration.

Satellite imagery showed Crane's terminus retreating by 6 km in the first month following the disintegration event (Fig. 2a-c), with iceberg calving known to be triggered by changes in the state and integrity of adjoining landfast sea ice (Christie et al., 2022; Arthur et al., 2021; Miles et al., 2017). Following the removal of the fast ice and melange from our model domain, an increase in extensional stresses was found in the vicinity of the terminus (Fig. 4), the magnitude of which mirrors that reported in the idealised model scenario of Surawy-Stepney et al. (2024). With these increases seen perpendicular to the terminus, it is

likely that the loss of buttressing from the fast ice and melange played a key role in the initial calving response following the fast ice disintegration (Benn et al., 2007; Walter et al., 2010). It is noted that calving may have occurred more readily at this time as the near-terminus area was characterised by susceptible, highly damaged ice (Fig. 2a), similar to observations reported by Amundson et al. (2010). A further assessment into the changes in stress regime following calving of the floating ice would be required to better understand the progressive calving behaviour seen later in the year.

As Crane's floating ice shelf also provided buttressing against upstream ice flow (Fig. 3), it follows that with each subsequent calving event, further resistive backstress was lost, ultimately leading to further acceleration of ice flow. With changes in grounding line flux most sensitive to loss of buttressing close to the location of the grounding line (Mitcham et al., 2022), we attribute any perceived delayed response in ice flow acceleration to be part of the evolving dynamic response of Crane following the disintegration of the fast ice.

## 5.3 Thickness of Fast Ice and Melange Elements

We configured the thickness of the melange in our model using a high resolution surface elevation profile (Howat et al., 2019), defined at 2m resolution with an associated $\pm 2$m error estimate per pixel. The average surface elevation of model elements downstream of the December 2021 terminus location is $4.57\,\mathrm{m}$ (41.66m thickness), with minimum and maximum surface elevations and standard deviation of $0.022\,\mathrm{m}$, $50.69\,\mathrm{m}$ and $7.33\,\mathrm{m}$ respectively. If it is assumed that the maximum associated error of 2m exists over the entire modelled area of melange, the upper limit to the average change in surface elevation in this area is 43.75%.

The thinnest configuration of melange that we tested considered a 40% reduction in surface elevation, which resulted in a 14.3% increase in the calculated mean buttressing ratio compared to the reference configuration (Fig. 5b). Results from the other sensitivity experiments were typically within about 6% of the reference configuration (Fig. 5) and the amount by which these sensitivities deviated from the reference case was uncorrelated to the percentage change in surface elevation. These results show that despite considerable uncertainties in the thickness of the fast ice and melange, the buttressing ratios calculated at the glacier terminus are not significantly impacted by these uncertainties. Though the rheology rate factor, $A$, is calculated in the inversion, it is the product of $A \cdot h$ that allows us to capture the stress regime throughout the melange.

Results from the sensitivity experiments support the conclusion that the melange buttressed the glacier prior to its disintegration, despite variation being seen in the distribution and magnitude of the buttressing ratios calculated at each sample point across the terminus between sensitivity cases. Mean buttressing ratios at the glacier terminus were between 0.67 and 0.72 in 7 out of the 8 sensitivity cases assessed, and below 0.8 in all sensitivity experiments.

## 6 Conclusions

We defined the stress regime in Crane glacier and the surrounding ambient fast ice and melange through inverse modelling, allowing us to quantify the buttressing provided to Crane by a realistic melange profile for the first time.

Our results show that Crane was buttressed by the melange prior to the disintegration of multi-year landfast sea ice over the Larsen B embayment in January 2022. A mean buttressing ratio of $\Theta_N = 0.68$ was found at the interface between the glacier and ice melange. Further, we have shown that the stress distribution over regions of melange can be modelled effectively by considering the product of the rheology rate factor, $A$, optimised through inversion of measured velocities, and the input ice thickness, $h$. Uncertainties in the thickness of the fast ice and melange therefore has little impact on the calculated butressing ratios.

Observations showed a 3.4% increase in flow speed at the ice front and the rapid loss of $6\,\mathrm{km}$ of Crane's floating ice shelf in the first month after the disintegration, whilst our model simulations found a perturbation in the near-terminus stress regime following the removal of melange from the model domain. An average increase in extensional stresses normal to the terminus of $19.2\,\mathrm{kPa}$ was found, which likely triggered the rapid calving of Crane's floating ice over a region which was already highly damaged. The consequent loss of large portions of the floating ice shelf led to further reduction in resistive backstresses to the glacier, which was followed by further acceleration of ice flow throughout 2022.

In order to gain a more comprehensive understanding of how significant the role of melange and fast ice may be in buttressing ice shelves and outlet glaciers, future research should focus on different regions where topography and flow characteristics

differ from that of Crane. The methodology of this study can be applied to any area of interest if good quality observational velocity fields of both the glacier and melange are available to constrain the model for accurate calculation.

*Code and data availability.* The open-source ice-flow model Úa used for the numerical simulations is preserved at https://doi.org/10.5281/zenodo.3706624. Velocity data used in this study is available from Enveo's CryoPortal https://cryoportal.enveo.at/. LandSat-8 and LandSat-9 images were obtained courtesy of the U.S. Geological Survey from https://earthexplorer.usgs.gov/. The Reference Elevation Model of Antarctica strip DEMs

were obtained from the Polar Geospatial Center Fridge web application, https://fridge.pgc.umn.edu/.

*Author contributions.* RP led the study, performed modelling work and processed results. SS and GHG initiated the study and contributed to discussion and interpretation of results. JW and TN provided monthly averaged ice flow velocity data for the study site. RP prepared the manuscript with contributions from all authors.

*Competing interests.* The authors declare that they have no conflict of interest.

*Acknowledgements.* Richard Parsons is supported by the National Environmental Research Council (NERC) funded ONE Planet Doctoral Training Partnership, NE/S007512/1, hosted by Northumbria and Newcastle Universities. Sainan Sun is funded by grant ISOTIPIC: NERC highlight topics 2023, NE/Z503344/1. G. Hilmar Gudmundsson was partially funded through grant NSFGEO-NERC:IceRift, NE/V013319/1. We acknowledge the use of datasets produced through the ESA project Antarctic Ice Sheet Climate Change Initiative (AIS CCI).

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
