# Peer review of "Quantifying the Buttressing Contribution of Landfast Sea Ice and Melange to Crane Glacier, Antarctic Peninsula"

_EGUsphere, 2024_

## Referee Comment (RC1)

*"Quantifying the Buttressing Contribution of Sea Ice to Crane Glacier",* **by Parsons et al. - egusphere-2024-1499**

The manuscript quantifies the buttressing effect of land-fast sea ice on the Crane Glacier. The authors use the 2022 disintegration of the fast ice that occupied the Larsen B embayment as a reference to quantify the buttressing. The main findings showed that the land-fast sea ice provided significant buttressing to the glacier with a mean buttressing number of 0.68 and that the loss of this buttressing led to increased extensional stresses and rapid calving, triggering further acceleration of ice flow.

This study provides new insights into the significance of fast ice for the structural stability of ice shelves and how the resistant stresses of the floating ice respond to the absence of surrounding fast ice. It quantifies this effect for the first time using numerical modelling.

This work is an important addition to the current scientific debate about the importance of sea ice (in this article land-fast sea ice) in stabilising ice shelves. The novelty lies in using numerical modelling for the first time to quantify the buttressing effect of fast-ice/melange.

The manuscript is well written and gives a good picture of the current knowledge of sea ice - ice shelf interaction/ glacier interaction, with a specific review of the recent work on the 2022 disintegration event.

**Comments regarding the overall text:**

One of the main issues that needs to be addressed before publication is the use of the term "sea ice." This term encompasses various types of ice. This article focuses on sea ice that is connected to the land (known as land-fast sea ice or fast-ice), and contains calving debris of different sizes, forming a dense melange. It's important to note that this type of "sea ice" is not smooth, nor even in its thickness, and this distinction is crucial because a reader might assume that any sea ice type could generate buttressing.

The paper would benefit if it included a more in-depth discussion about the minimum expected thickness that would generate significant backstress. According to the sensitivity analysis, even a 40% reduction in this particular case won't make a significant change. This is important because it relates to the likely thickness that fast-ice might need to create buttressing, and this will depend on the age and genesis of the formation of the fast-ice/melange.

Please make sure to clarify the methodology used to define the grounding line. It was mentioned that the floating criteria is used in combination with the REMA DEM, but it's important to provide more details. Additionally, has this method been compared with other datasets, such as the one described in the study by Rott et al. (2018)? In the supplementary materials this is mentioned with more detail. Please add the references from where the grounding line was taken to the main text.

Is there space to create a map similar to the ones that Fürst et al. (2016) created for Antarctica, but for the entire boundary area? Additionally, it would be beneficial to include a

paragraph in the conclusions discussing how further research in other regions of Antarctica, which are not fjord-like embayments, will contribute to a comprehensive understanding of the significance of melange/land-fast sea ice in buttressing Antarctic glaciers/ice shelves.

If section 3.6 it's joined with 4.3 the sensitivity analysis reads better and it's easier to understand. As it is now it reads disjointed, with 3.6 finishing abruptly. Lastly, please ensure that all the supplementary materials are well formatted and the captions are correct.

**Comments on figures:**

For ALL figures, the size of ticks and axes labels must be larger. They should be easy to read on an A4 printed paper.

Figure 1: The colours selected for the model boundary, glacier and fast-ice extent (filled and dashed) lines are hard to see, especially the dashed ones; choose colours with better contrast over the underlying image, and consider that it might be printed in greyscale. Please add labels to the different panes (A and B) and refer to them in the text accordingly. Add the glacier terminus over the Dec-2021 image. Could it be possible to add a panel showing just the terminus of the glacier? In order to understand how fractured the glacier is and how the terminus was defined. Mention in the legend that the image is in polar stereographic projection.

Figure 2: Please add the dates of acquisition on the different panel titles and labels to each panel (a, b, c, d, and e) and use them when referencing in the text. If the projection is added, there is no need to add the x and y labels. I will leave the ticks labels in metres and less frequently, the same for Figures 1, 3e, 4, and 5. I would add the word "digitised" to: …the white line shows the digitised terminus position.

Figure 3: Even though "dotted line" is correct when referring to a dotted or dashed line, I would specifically write "dashed line" for the different examples shown here. Increase the size of the month-year labels in panel e.

Figure 4: An inset zooming into the edge of the terminus on panel b would help to better visualize the exchange in stress distribution at the terminus. The blue arrows are barely visible; the inset would also help with this. The last sentence can be removed as is also in the text.

[Figure]

Figure 5: This figure is important for understanding the thickness distribution along the glacier margins and its relationship to the buttressing number.However, it's quite challenging to comprehend this from the surface models. To help with this, it would be beneficial to have a graph showing the thickness distribution and another graph displaying the variability of the buttressing number along the sampled line. Furthermore, including only two end members and the original DEM, in addition to the graphs, will help focus on highlighting the low significance of the thickness reduction on the buttressing number.

**In-text comments:**

Line 14 to 20 - Introduce Fraser, et al 2021 and/or Fraser et al 2023, to present changes in fast ice that are more relatable to this paper than changes in broader sea ice.

Line 25 - It says … of sea ice … it should say … of land-fast sea ice …, or fast -ice if it was defined before.

Line 65 - add Gudmundsson, 2013 alongside Schoof, 2007.

Line 85 - Floatation Criterion should be Flotation Criterion.

Line 85 - Citations are needed for the grounding line and floatation criterion.

Line 100 to 104 - Could this long sentence be divided in two? It's very hard to comprehend what the authors are trying to say.

Line 220 - It says … the sea ice … it should say … the fast-ice …, or melange.

Line 265 - Please add more statistics, standard deviation, min, max of the ice/melange thickness.

Line 275 - Again not sea ice, either fast-ice or melange, please change in other places throughout the text.

**References**

Fraser, A. D., Wongpan, P., Langhorne, P. J., Klekociuk, A. R., Kusahara, K., Lannuzel, D., ... & Wienecke, B. (2023). Antarctic landfast sea ice: A review of its physics, biogeochemistry and ecology. Reviews of Geophysics, 61(2), e2022RG000770.

Fraser, A. D., Massom, R. A., Handcock, M. S., Reid, P., Ohshima, K. I., Raphael, M. N., ... & Porter-Smith, R. (2021). Eighteen-year record of circum-Antarctic landfast-sea-ice distribution allows detailed baseline characterisation and reveals trends and variability. The Cryosphere, 15(11), 5061-5077.

Fürst, J. J., Durand, G., Gillet-Chaulet, F., Tavard, L., Rankl, M., Braun, M., & Gagliardini, O. (2016). The safety band of Antarctic ice shelves. Nature Climate Change, 6(5), 479-482.

Gudmundsson, G.: Ice-shelf buttressing and the stability of marine ice sheets, The Cryosphere, 7, 647–655, 2013

Rott, H., Abdel Jaber, W., Wuite, J., Scheiblauer, S., Floricioiu, D., Van Wessem, J. M., ... & Van Den Broeke, M. R. (2018). Changing pattern of ice flow and mass balance for glaciers discharging into the Larsen A and B embayments, Antarctic Peninsula, 2011 to 2016. The Cryosphere, 12(4), 1273-1291.

---

## Referee Comment (RC2)

**Review of "Quantifying the Buttressing Contribution of Sea Ice to Crane Glacier" by Parsons et al., 2024.**

This article deals with the interesting and important topic of how the presence of mélange can affect viscous stresses within ocean-terminating glaciers. The authors focus on the calving and acceleration of Crane Glacier following the removal of landfast sea ice from the Larsen-B Embayment in early 2022. They use a numerical model to calculate a local "buttressing number" representing the amount of buttressing provided to Crane Glacier by proglacial mélange prior to its removal, and changes in viscous stress within the glacier following its removal. They find the removal of mélange caused perturbations in stress of order 10kPa and suggest that this caused the retreat and acceleration of Crane Glacier from 2022 onwards.

This study is well motivated, scientifically sound and makes a robust and important contribution to an interesting discussion that has been the subject of recent literature. The article is well written and should be published in The Cryosphere as it will be valued by a broad range of cryospheric scientists. However, this is subject to some revisions - particularly regarding the use of the words "sea ice", the placement of the research in the context of other recent literature, and the general quality of the figures.

The following review gives some general comments on the article before listing a set of additional, specific comments that are not covered there.

**General comments:**

In a number of places throughout the article, the authors use the term "sea ice" where I think they mean "ice mélange" (including in the title of the article). This distinction is both important to how the reader should interpret the results of the article, and how to place the work within the context of contemporary literature (see below). Though sea ice is an important component of mélange, it is a quite different material with different properties and conclusions regarding the ability of mélange to bear stress shouldn't necessarily be applied to pure sea ice (even when it's thick). This should not be too difficult to fix throughout the article and will improve it significantly.

The use of the term "sea ice" also slightly muddies the waters regarding previous work on this area. In general, though the authors do a nice job of summarising previous work on landfast sea ice in the Larsen-B Embayment, the assessment of how previous modelling work relates to this study should be improved. For example, there are a number of references to this being the first example of using a numerical model to calculate the effect that landfast sea ice has on englacial stress, but Surawy-Stepney et al., 2024 (who used a modelling set-up which explicitly neglected the presence of mélange) calculate an upper limit for this in their figure 4d. Looking at that figure, the numbers they come up with are the same order of magnitude as those presented here (10s of kPa). In which case, the biggest differences between the two studies seem to be that SS24 consider the idealised case of pure landfast sea ice, while this study considers the realistic case of proglacial mélange, and in the interpretation of the resulting stress changes as 'big' or 'small'. The reader could be left with the impression that this article disagrees fundamentally with previous modelling efforts, whereas in reality there seems to be no contradiction (e.g. not one arising from the different methods of initialising the rate factor prior-to or post removal of proglacial ice, as suggested in the introduction).

The figures are nicely placed throughout the article and cover all the necessary bases, but they are a little bit difficult to read in general. Making the text quite a bit bigger in the figures would go a long way to fixing the issue.

**Specific comments:**

Line 76: Could the authors elaborate on how the DEM strips were co-registered?

Line 116: Out of curiosity, could the authors elaborate on these boundary conditions a bit? The use of a boundary condition on speed seems sensible given the domain doesn't reach the edges of the fjord or the margin of the landfast sea ice. Is it the case that stress boundary conditions would be more desirable given that it's the stress across the domain that we're ultimately interested in and we don't know the mélange rheology? Is the speed used to define the boundary conditions the same as that used in the inverse problem? Does it matter that this data is probably noisy?

Line 128: The use of "prior" makes the reader think of prior distributions in the Bayesian sense which I don't think would be right unless the regularisation terms have a particular form (and these values are the mean of the prior distribution). If so, it might be better to say "initial guesses for A and C" or something to avoid confusion.

Line 130: Tikhonov regularisation is a class of regularisation methods, which one in particular was used?

Section 3.3: It would be great to see (perhaps just as a supplementary figure) maps of: 1) ice speed used in the inverse problem, 2) the rate factor found at convergence, and 3) the solution misfit. This would help the reader gauge the success of the inverse problem (which I'm sure is very good) particularly regarding transition across the calving front. E.g. I would be interested to see whether the ice speed data is any noisier in the proglacial mélange than on the ice shelf.

Line 148: It might be worth making it clear that these are vertically-averaged stresses to avoid any confusion.

Section 4.1: Given that this is the first attempt to quantify buttressing using the buttressing number, it would be nice to see some more discussion of the sources of uncertainty associated with the numbers computed. Even if exact quantification is difficult, how large might these uncertainties be expected to be? The authors' exploration of solutions to the inverse problem with different mélange thickness, and how the model compensates by changing the rate factor to produce similar stress fields, is nice. However, there are likely to be many sources of uncertainty and it's not clear to me how many of these are covered by the thickness sensitivity experiments.

Figure 4: A key showing how the lengths of the lines corresponds to vertically-averaged stress would be great.

Discussion/section 4.2: It would be nice to see some discussion about how these numbers compare with those computed in Surawy-Stepney et al., (2024) – particularly their figure 4. To me they seem fairly similar, perhaps their stiff-but-thin sea ice has a similar load-bearing capacity to your weak-but-thick mélange?

Section 5.2: For those who find it a bit difficult to contextualise numbers such as "stress of 60kPa" or "change in stress of 19.2kPa" it might be helpful to have a reference stress to consider. For example, I imagine basal stresses calculated during the inverse problem are on the order 100kPa? This section might be a nice place to put a sentence with this comparison.

Lines 290 onward: The conclusion suggests that inverting for the rate factor again might allow for the inclusion of sea ice of unknown thickness/rheology in numerical models. It is worth noting that it is possible to get away with this in diagnostic simulations such as that presented here, and when considering mélange. However, though the stress distribution found using this kind of method during

the inverse problem might be roughly correct, transient simulations will require a better treatment of sea ice/mélange rheology, which differs from that of meteoric ice by more than just its viscosity!

**Editorial comments:**

Line 22: Gudmundsson reference should be at the end.

Equation 6: That should be $2\tau_{yy} + \tau_{xx}$

Line 203: "lead" should be "led".

Figure 2: Should the last line of the caption read "solid black line" rather than "dashed yellow line"?

**References:**

Surawy-Stepney, T., Hogg, A. E., Cornford, S. L., Wallis, B. J., Davison, B. J., Selley, H. L., Slater, R. A. W., Lie, E. K., Jakob, L., Ridout, A., Gourmelen, N., Freer, B. I. D., Wilson, S. F., and Shepherd, A.: The effect of landfast sea ice buttressing on ice dynamic speedup in the Larsen B embayment, Antarctica, The Cryosphere, 18, 977–993, https://doi.org/10.5194/tc-18-977-2024, 2024.

---

## Author Comment (AC2)

Authors' Response to Review of **egusphere-2024-1499 - *"Quantifying the Buttressing Contribution of Sea Ice to Crane Glacier"***

We wish to express appreciation to the Reviewers for their insightful comments, which have helped us to significantly improve our manuscript. According to the suggestions, we revised our manuscript, which is enclosed.

Following the reviewers' suggestions, we have made some general changes to the manuscript, including updating the use of the term 'sea ice' to more specifically refer to land fast sea ice and ice melange. We have also updated all of our figures to make reading and interpretation of results easier. Specific changes to the manuscript are detailed in our responses to each of the Reviewer comments below.

**Reviewer #1**

The manuscript quantifies the buttressing effect of land-fast sea ice on the Crane Glacier. The authors use the 2022 disintegration of the fast ice that occupied the Larsen B embayment as a reference to quantify the buttressing. The main findings showed that the land-fast sea ice provided significant buttressing to the glacier with a mean buttressing number of 0.68 and that the loss of this buttressing led to increased extensional stresses and rapid calving, triggering further acceleration of ice flow.

This study provides new insights into the significance of fast ice for the structural stability of ice shelves and how the resistant stresses of the floating ice respond to the absence of surrounding fast ice. It quantifies this effect for the first time using numerical modelling.

This work is an important addition to the current scientific debate about the importance of sea ice (in this article land-fast sea ice) in stabilising ice shelves. The novelty lies in using numerical modelling for the first time to quantify the buttressing effect of fast-ice/melange.

The manuscript is well written and gives a good picture of the current knowledge of sea ice - ice shelf interaction/ glacier interaction, with a specific review of the recent work on the 2022 disintegration event.

We thank the reviewer for the careful review and recognising the importance of our work.

**General Comments**

One of the main issues that needs to be addressed before publication is the use of the term "sea ice." This term encompasses various types of ice. This article focuses on sea ice that is connected to the land (known as land-fast sea ice or fast-ice), and contains calving debris of different sizes, forming a dense melange. It's important to note that this type of "sea ice" is not smooth, nor even in its thickness, and this distinction is crucial because a reader might assume that any sea ice type could generate buttressing.

In the revised version of the manuscript, we have replaced the use of sea ice throughout the document and been explicit in our reference to landfast sea ice and the pro-glacial melange. The title of the manuscript has also been updated to reflect this correction in terminology.

The paper would benefit if it included a more in-depth discussion about the minimum expected thickness that would generate significant backstress. According to the sensitivity analysis, even a 40% reduction in this particular case won't make a significant change. This is important

because it relates to the likely thickness that fast-ice might need to create buttressing, and this will depend on the age and genesis of the formation of the fast-ice/melange.

We agree with the reviewer that the melange thickness is an important index to calculate buttressing. However, it's hard to determine from our study the minimum thickness that could buttress the glacier. This is because the quantity of buttressing supplied by the melange relies on both ice rheology and ice thickness. The inverse method assures the overall buttressing effect is robust and not sensitive to the unknown melange ice thickness, which is the purpose of the ice thickness sensitivity study.

Please make sure to clarify the methodology used to define the grounding line. It was mentioned that the floating criteria is used in combination with the REMA DEM, but it's important to provide more details. Additionally, has this method been compared with other datasets, such as the one described in the study by Rott et al. (2018)? In the supplementary materials this is mentioned with more detail. Please add the references from where the grounding line was taken to the main text.

Using the flotation criterion, we calculate the submerged thickness of ice based on the surface elevation considering assumed constant ice and ocean densities of 917 kg m$^3$ and 1030kg m$^3$ respectively. The grounding line is then found in our model where this submerged thickness reaches the bedrock. This further clarification has been added in the revised manuscript. As the bed topography is poorly constrained in this region, we are presented with a choice of possible bed topographies from previously published datasets of bed elevation which in turn affect the computed grounding line location. The sensitivity experiment presented in the supplementary data was performed to show that the results and conclusions of the study are robust in the absence of accurate knowledge of the bed topography and grounding line location at the time of the fast ice disintegration. We show here that considering different bed topography and consequently different grounding line locations, the conclusions of the work remain unchanged. We provide context for our computed grounding line locations through comparison with a contemporary grounding line product for the Antarctic Peninsula (Wallis et al, 2024). We have kept this discussion in the supplementary data.

Is there space to create a map similar to the ones that Fürst et al. (2016) created for Antarctica, but for the entire boundary area? Additionally, it would be beneficial to include a paragraph in the conclusions discussing how further research in other regions of Antarctica, which are not fjord-like embayments, will contribute to a comprehensive understanding of the significance of melange/land-fast sea ice in buttressing Antarctic glaciers/ice shelves.

This is an insightful point for future research. The method from this study can be applied to any glaciers of interest. As we found that the melange in Larsen B is buttressing the glacier, it's sensible to explore other ice shelves in Antarctica. One of the essential requirements is to have good quality observational velocity fields of both the glacier and melange to constraint the model for accurate calculation. We added the following discussion in the conclusion: '*In order to gain a more comprehensive understanding of the significance of melange and landfast sea ice in buttressing ice shelves and outlet glaciers, future research should focus on different regions where topography and flow characteristics differ from that of Crane. The method from this study can be applied to any area of interest if good quality observational velocity fields of both the glacier and melange are available to constrain the model for accurate calculation.*'

If section 3.6 it's joined with 4.3 the sensitivity analysis reads better and it's easier to understand. As it is now it reads disjointed, with 3.6 finishing abruptly. Lastly, please ensure that all the supplementary materials are well formatted and the captions are correct.

Sections 3.6 and 4.3 have been left separate in order to describe all methodologies employed in this study prior to the results section. However, we have added an additional sentence to the end of Section 3.6 to link this with the results section. As per the comments below on the figures in general, the supplementary figures have also been revised reflecting these comments along with updates to captions.

**Comments on figures**

For ALL figures, the size of ticks and axes labels must be larger. They should be easy to read on an A4 printed paper.

The sizing of ticks, axes and labels in each figure has been revised in order to make reading and interpretation easier.

Figure 1: The colours selected for the model boundary, glacier and fast-ice extent (filled and dashed) lines are hard to see, especially the dashed ones; choose colours with better contrast over the underlying image, and consider that it might be printed in greyscale. Please add labels to the different panes (A and B) and refer to them in the text accordingly. Add the glacier terminus over the Dec-2021 image. Could it be possible to add a panel showing just the terminus of the glacier? In order to understand how fractured the glacier is and how the terminus was defined. Mention in the legend that the image is in polar stereographic projection.

The colours used for the domain boundary and ice extents have now been updated and panel labels have been added. As a close up of the terminus area is given in figure 2, this is not included again within this figure. Similarly, the digitised terminus location is shown in figure 2. The caption has been updated to mention that the image is in polar stereographic projection. Latitude and longitude labels have also been added to panel a) to aid interpretation.

Figure 2: Please add the dates of acquisition on the different panel titles and labels to each panel (a, b, c, d, and e) and use them when referencing in the text. If the projection is added, there is no need to add the x and y labels. I will leave the ticks labels in metres and less frequently, the same for Figures 1, 3e, 4, and 5. I would add the word "digitised" to: ...the white line shows the digitised terminus position.

As the terminus dates are discussed throughout the manuscript, we have included details of satellite and acquisition dates in Table 1, which is referenced in the figure caption and throughout the text. We keep this additional information in the table in order to keep text concise. We have added panel labels, which have also now been specifically referred to in the text for clarity. It is our preference to keep tick labels in kilometres but appreciate there will be a difference in opinion from reader to reader. The caption has been updated to state 'digitised terminus location'.

Figure 3: Even though "dotted line" is correct when referring to a dotted or dashed line, I would specifically write "dashed line" for the different examples shown here. Increase the size of the month-year labels in panel e.

We have updated to state 'dashed-line' and font sizes have been increased throughout the figure.

Figure 4: An inset zooming into the edge of the terminus on panel b would help to better visualize the exchange in stress distribution at the terminus. The blue arrows are barely visible; the inset would also help with this. The last sentence can be removed as is also in the text.

This figure has been updated to include a zoomed in view close to the terminus to visualise both the absolute stresses with the fast ice intact and the change in stress distribution following removal of fast ice in the diagnostic run. Further updates include increases to font sizing throughout as well as a key to illustrate the magnitude of stresses compared to arrow size.

Figure 5: This figure is important for understanding the thickness distribution along the glacier margins and its relationship to the buttressing number. However, it's quite challenging to comprehend this from the surface models. To help with this, it would be beneficial to have a graph showing the thickness distribution and another graph displaying the variability of the buttressing number along the sampled line. Furthermore, including only two end members and the original DEM, in addition to the graphs, will help focus on highlighting the low significance of the thickness reduction on the buttressing number.

We have updated Figure 5 to aid the interpretation of results from this figure. Updates include displaying a greater area of the melange region for which the surface elevation profiles are adjusted in the sensitivity study in order to better illustrate the overall changes of the ice thickness distributions. We have also corrected the mean buttressing ratio which is stated on the reference case (panel A) to 0.68 which was previously incorrectly stated as 0.72 due to an error in our processing script. We have kept plots for each sensitivity case to show variability between cases, particularly as this does not necessarily occur linearly. All labels, ticks and fonts have been increased in size.

**In-text comments**

Line 14 to 20 - Introduce Fraser, et al 2021 and/or Fraser et al 2023, to present changes in fast ice that are more relatable to this paper than changes in broader sea ice.

The introduction has been updated with more specific reference to landfast sea ice rather than a broader reference to sea ice. The additional suggested citations have been incorporated.

Line 25 - It says ... of sea ice ... it should say ... of land-fast sea ice ..., or fast -ice if it was defined before.

Mentions of sea ice have been updated to refer to landfast sea ice and ice melange throughout.

Line 65 - add Gudmundsson, 2013 alongside Schoof, 2007.

This citation has been added

Line 85 - Floatation Criterion should be Flotation Criterion.

This has been corrected

Line 85 - Citations are needed for the grounding line and floatation criterion.

Grounding line locations in the ice sheet model are calculated based on geometry and Archimedes principle. We added citations of the datasets.

Line 100 to 104 - Could this long sentence be divided in two? It's very hard to comprehend what the authors are trying to say.

These lines have been re-written as follows: "*These terminus locations were considered in order to compare the buttressing stresses provided by the fast and melange with those provided by the portions of Crane's floating ice shelf which were lost to calving in the months following the fast ice disintegration.*"

Line 220 - It says ... the sea ice ... it should say ... the fast-ice ..., or melange.

Mentions of sea ice have been updated to refer to landfast sea ice and ice melange throughout.

Line 265 - Please add more statistics, standard deviation, min, max of the ice/melange thickness.

Minimum, maximum surface elevations and standard deviation of 0.022m 50.69m and 7.33m respectively have been added to the discussion.

Line 275 - Again not sea ice, either fast-ice or melange, please change in other places throughout the text.

Mentions of sea ice have been updated to refer to landfast sea ice and ice melange throughout.

**Reviewer #2**

This article deals with the interesting and important topic of how the presence of mélange can affect viscous stresses within ocean-terminating glaciers. The authors focus on the calving and acceleration of Crane Glacier following the removal of landfast sea ice from the Larsen-B Embayment in early 2022. They use a numerical model to calculate a local "buttressing number" representing the amount of buttressing provided to Crane Glacier by proglacial mélange prior to its removal, and changes in viscous stress within the glacier following its removal. They find the removal of mélange caused perturbations in stress of order 10kPa and suggest that this caused the retreat and acceleration of Crane Glacier from 2022 onwards.

This study is well motivated, scientifically sound and makes a robust and important contribution to an interesting discussion that has been the subject of recent literature. The article is well written and should be published in The Cryosphere as it will be valued by a broad range of cryospheric scientists. However, this is subject to some revisions - particularly regarding the use of the words "sea ice", the placement of the research in the context of other recent literature, and the general quality of the figures.

The following review gives some general comments on the article before listing a set of additional, specific comments that are not covered there.

We thank the reviewer for the positive comments and constructive suggestions.

**General Comments**

In a number of places throughout the article, the authors use the term "sea ice" where I think they mean "ice mélange" (including in the title of the article). This distinction is both important to how the reader should interpret the results of the article, and how to place the work within the context of contemporary literature (see below). Though sea ice is an important component of mélange, it is a quite different material with different properties and conclusions regarding the ability of mélange to bear stress shouldn't necessarily be applied to pure sea ice (even when it's thick). This should not be too difficult to fix throughout the article and will improve it significantly.

In the revised version of the manuscript, we have replaced the use of sea ice throughout the document and been explicit in our reference to landfast sea ice and the pro-glacial melange. The title of the manuscript has also been updated to reflect this correction in terminology.

The use of the term "sea ice" also slightly muddies the waters regarding previous work on this area. In general, though the authors do a nice job of summarising previous work on landfast sea ice in the Larsen-B Embayment, the assessment of how previous modelling work relates to this study should be improved. For example, there are a number of references to this being the first example of using a numerical model to calculate the effect that landfast sea ice has on englacial stress, but Surawy-Stepney et al., 2024 (who used a modelling set-up which explicitly neglected the presence of mélange) calculate an upper limit for this in their figure 4d. Looking at that figure, the numbers they come up with are the same order of magnitude as those presented here (10s of kPa). In which case, the biggest differences between the two studies seem to be that SS24 consider the idealised case of pure landfast sea ice, while this study considers the

realistic case of proglacial mélange, and in the interpretation of the resulting stress changes as 'big' or 'small'. The reader could be left with the impression that this article disagrees fundamentally with previous modelling efforts, whereas in reality there seems to be no contradiction (e.g. not one arising from the different methods of initialising the rate factor prior-to or post removal of proglacial ice, as suggested in the introduction).

Our intention was to flag the novelty of this study which was the inclusion of the realistic geometry of the adjoining melange elements and inversion over this region to define its rheology and consequently the associated backstresses. We in no way mean to detract from the work done by Surawy-Stepney using an idealised setup. We have been more careful to make this distinction in the revised manuscript and have added interpretation of results in context with the findings of Surawy-Stepney et al (2024), as the findings of both studies, despite differing methodologies, point towards similar conclusions.

The figures are nicely placed throughout the article and cover all the necessary bases, but they are a little bit difficult to read in general. Making the text quite a bit bigger in the figures would go a long way to fixing the issue.

The sizing of ticks, axes and labels in each figure has been revised in order to make reading and interpretation easier.

**Specific Comments**

Line 76: Could the authors elaborate on how the DEM strips were co-registered?

This section has been updated as follows: "*In order to configure a model geometry representative of Crane Glacier prior to the disintegration of fast ice in the Larsen B embayment, we utilised a range of REMA strip DEMs, (Howat et al., 2019) timestamped between 30th September 2020 and 16th January 2022. The strips are defined at 2 m spatial resolution and have absolute error of +/-2m in horizontal and vertical planes. To create a continuous surface elevation profile across our model domain, each strip was added in turn and where coordinates overlapped between strips, the most recent strip was used. As elevations at intersecting locations did not deviate beyond the associated error range, no further corrections were applied between intersecting strips. The near-terminus region and melange filled outlet of Crane were populated by the strip dated 16th January 2022.*"

Line 116: Out of curiosity, could the authors elaborate on these boundary conditions a bit? The use of a boundary condition on speed seems sensible given the domain doesn't reach the edges of the fjord or the margin of the landfast sea ice. Is it the case that stress boundary conditions would be more desirable given that it's the stress across the domain that we're ultimately interested in and we don't know the mélange rheology? Is the speed used to define the boundary conditions the same as that used in the inverse problem? Does it matter that this data is probably noisy?

We apply hydrostatic pressure at the ocean terminating boundary, and zero flow conditions for the interior boundaries. The extent of the computational boundary (Figure 1) includes the fjord, therefore the ice melange and landfast sea ice are included in the model. As the ice front is not the computational boundary, we don't need to apply any boundary condition there. When we solve the SSA approximation of the momentum equation in the ice-sheet model, we calculate the horizontal velocity fields using vertically integrated rate factor (A*h), and we estimate rate factor A in our inverse runs. As long as we could reproduce observational velocity for both the

glacier and the melange, the stress field at the calving front (hence the buttressing stresses of downstream melange) should be realistic. The advantage of this setup is that we don't make assumptions about the melange rheology, and the uncertainty from melange thickness is compensated by the rate factor A during inversion, to assure the correct A*h and back stress. This does mean that the accuracy of the back stress and buttressing towards the ice front is constrained by the noise in the observational velocity datasets. While observational datasets of the melange area can be very noisy, we selected monthly averaged velocity data close to the time of the fast ice disintegration whilst also seeking to minimise error associated with the velocity datasets. Sensitivity to the chosen velocity dataset is discussed in the supplementary information.

Line 128: The use of "prior" makes the reader think of prior distributions in the Bayesian sense which I don't think would be right unless the regularisation terms have a particular form (and these values are the mean of the prior distribution). If so, it might be better to say "initial guesses for A and C" or something to avoid confusion.

The priors are assigned ensuring they have a physical basis. They are indeed used in the regularisation terms to restrict overfitting. We have added further details of the form of regularisation in Section 3.3.

Line 130: Tikhonov regularisation is a class of regularisation methods, which one in particular was used?

The form that the regularisation takes has been added in section 3.3.

Section 3.3: It would be great to see (perhaps just as a supplementary figure) maps of: 1) ice speed used in the inverse problem, 2) the rate factor found at convergence, and 3) the solution misfit. This would help the reader gauge the success of the inverse problem (which I'm sure is very good) particularly regarding transition across the calving front. E.g. I would be interested to see whether the ice speed data is any noisier in the proglacial mélange than on the ice shelf.

Additional figures have been added to the supplementary data.

Line 148: It might be worth making it clear that these are vertically-averaged stresses to avoid any confusion.

In the revised manuscript we have stated that these are vertically integrated in order to avoid any confusion

Section 4.1: Given that this is the first attempt to quantify buttressing using the buttressing number, it would be nice to see some more discussion of the sources of uncertainty associated with the numbers computed. Even if exact quantification is difficult, how large might these uncertainties be expected to be? The authors' exploration of solutions to the inverse problem with different mélange thickness, and how the model compensates by changing the rate factor to produce similar stress fields, is nice. However, there are likely to be many sources of uncertainty and it's not clear to me how many of these are covered by the thickness sensitivity experiments.

Uncertainties could come from noise in observational datasets (e.g velocity field of ice melange and the ice shelf, the exact location of ice front, ice geometry), and caused numerically, however it is difficult to quantify the magnitude of these uncertainties. The thickness sensitivity experiments demonstrate the robustness of the inverse experiments, and that the resulting

buttressing ratio is independent of melange thickness. We added additional sources of uncertainty in a new paragraph in the discussion section 5.1: "*The buttressing numbers presented above are subject to a degree of uncertainty due to the difficulty in defining the exact location where the transition between glacier and melange exists. By assessing the normal resistive stresses at 100m intervals along the defined terminus location, we aimed to minimise this uncertainty by capturing the variability in ice thickness and rheology that may exist between melange elements. We have not tested the impact on the results that would be caused by changes to the defined terminus coordinates, however similar variability in the thickness and rheology between melange elements would be expected if the terminus location was moved elsewhere in the transition zone between the damaged ice shelf and melange region.*

*We note that further uncertainty may exist, arising from uncertainties in the datasets used in this study. We tested the sensitivity of results to the bed topography considered in the model geometry (Fig S4) and also to the velocity field considered in the inversion (Fig S6) to demonstrate the robustness of these results. We discuss the sensitivity of the results to the prescribed thickness of the fast ice and melange in Section 5.3.*"

Figure 4: A key showing how the lengths of the lines corresponds to vertically-averaged stress would be great.

A reference length scale has been added to each panel in the revised figure to illustrate the magnitude of stresses compared to arrow size.

Discussion/section 4.2: It would be nice to see some discussion about how these numbers compare with those computed in Surawy-Stepney et al., (2024) – particularly their figure 4. To me they seem fairly similar, perhaps their stiff-but-thin sea ice has a similar load-bearing capacity to your weak-but-thick mélange?

Additional text has been added to the discussion in the revised manuscript around the similarity in the magnitude of stress changes found between studies. Where our conclusions differ appears to be in the interpretation of how this stress perturbation relates to the observed calving response and in the manuscript we discuss the significance of the increased extensional stresses running perpendicular to the terminus.

Section 5.2: For those who find it a bit difficult to contextualise numbers such as "stress of 60kPa" or "change in stress of 19.2kPa" it might be helpful to have a reference stress to consider. For example, I imagine basal stresses calculated during the inverse problem are on the order 100kPa? This section might be a nice place to put a sentence with this comparison.

We have added the average basal drag approaching the grounding line as a reference and added a sentence to Section 4.2 to state this: ".. maximum and mean modelled increases in extensional stresses were found to be 70.8 kPa and 19.2 kPa respectively. *These values can be compared to the modelled basal drag values approaching the grounding line which have an average of 64.4kPa.*"

Lines 290 onward: The conclusion suggests that inverting for the rate factor again might allow for the inclusion of sea ice of unknown thickness/rheology in numerical models. It is worth noting that it is possible to get away with this in diagnostic simulations such as that presented here, and when considering mélange. However, though the stress distribution found using this kind of method during the inverse problem might be roughly correct, transient simulations will

require a better treatment of sea ice/mélange rheology, which differs from that of meteoric ice by more than just its viscosity!

We agree with the reviewer, that the approach in our study is appropriate for estimating buttressing effect offered by ice melange at time slots with observational data. The configuration can't be applied to transient runs, where both melange thickness and viscosity evolve with time. To do a transient simulation will require more input of melange thickness and viscosity, and probably additional physics (e.g. plastic deformation, interaction with ocean). Our study may motivate research on including melange in ice sheet models, but this is beyond the scope of this study. We have removed the ambiguous sentences from the final paragraph.

**Editorial Comments**

Line 22: Gudmundsson reference should be at the end.

Updated

Equation 6: That should be $2\tau yy + \tau xx$

We have corrected this equation

Line 203: "lead" should be "led".

Updated

Figure 2: Should the last line of the caption read "solid black line" rather than "dashed yellow line"?

The caption has been updated as pointed out in Figure 5 (rather than Figure 2)